# Deep learning mutation prediction enables early stage lung cancer detection in liquid biopsy

**Steven T. Kothen-Hill**
Weill Cornell Medicine, Meyer Cancer Center, New York, NY 10065
{sth2022}@med.cornell.edu

**Asaf Zviran, Rafi Schulman, Dillon Maloney, Kevin Y. Huang, Will Liao, Nicolas Robine**
New York Genome Center, New York, NY 10003, USA
{azviran,rschulman,dmaloney,khuang,wliao,nrobine}@nygenome.org

**Sunil Deochand**
New York University, New York, NY 10003
{sdd325}@nyu.edu

**Nathaniel D. Omans**
Tri-Institutional Program in Computational Biology and Medicine, Weill Cornell Medicine, New York, NY 10065
{nao2013}@med.cornell.edu

**Dan A. Landau**
Weill Cornell Medicine, Division of Hematology and Medical Oncology, New York, NY 10065
{dal3005}@med.cornell.edu

## Abstract

Somatic cancer mutation detection at ultra-low variant allele frequencies (VAFs) is an unmet challenge that is intractable with current state-of-the-art mutation calling methods. Specifically, the limit of VAF detection is closely related to the depth of coverage due to the requirement of multiple supporting reads in extant methods, precluding the detection of mutations at VAFs that are orders of magnitude lower than the depth of coverage. Nevertheless, the ability to detect cancer-associated mutations in ultra low VAFs is a fundamental requirement for low-tumor burden cancer diagnostics applications such as early detection, monitoring, and therapy nomination using liquid biopsy methods (cell-free DNA). Here we defined a spatial representation of sequencing information adapted for convolutional architecture that enables variant detection in a manner independent of the depth of sequencing. This method enables the detection of cancer mutations even in VAFs as low as $10^{-4}$, more than two orders of magnitude below the current state-of-the-art. We validated our method on both simulated plasma and on clinical cfDNA plasma samples from cancer patients and non-cancer controls. This method introduces a new domain within bioinformatics and personalized medicine - somatic whole genome mutation calling for liquid biopsy.

## 1 Introduction

The cancer genome acquires somatic mutations which drive its proliferative capacity (Lawrence et al., 2014). Mutations in the cancer genome also provide critical information regarding the evolutionary history and mutational processes active in each cancer (Martincorena et al., 2017; Alexandrov et al., 2013). Cancer mutation calling in patient tumor biopsies has become a pivotal step in determining patient outcomes and nomination of personalized therapeutics.

Identifying cancer mutations in liquid biopsy techniques, such as cell-free circulating DNA (cfDNA), has been suggested as a transformative platform for early-stage cancer screening and residual disease monitoring. cfDNA released from dying tumor cells enables surveys the somatic genome dynamically over time for clinical purposes, empowered by the ability to obtain cancer-related genetic material non-invasively through a simple blood draw. Circulating tumor DNA (ctDNA) can be found and measured in the plasma cfDNA of cancer patients. ctDNA was shown to correlate with tumor burden and change in response to treatment or surgery (Diehl et al., 2008). For example, ctDNA can be detected even in early stage non-small cell lung cancer (NSCLC) and therefore has the potential to transform NSCLC diagnosis and treatment (Sozzi et al., 2003; Tie et al., 2016; Bettegowda et al., 2014; Wang et al., 2010). Nevertheless, the fraction of ctDNA of the total cfDNA is typically exceedingly low, especially in low disease-burden contexts such as early detection or detection of residual disease after therapeutic interventions. While detection of cancer through cfDNA in the low disease-burden setting may be of significant clinical benefit, it challenges our current methods for identifying somatic mutations due to the ultra-low VAFs compared with the available depth of sequencing.

The most common type of somatic mutations is single-nucleotide variants (SNVs), which occur at a frequency of 1-100 per million bases. These variants are typically identified in sequencing data through a careful comparison of the DNA sequencing reads which map to a particular genomic locus in both the cancer DNA and the matched germline DNA. This process has been enabled through tools of ever-increasing sophistication that refine the statistical comparison between the number of reads supporting a candidate mutated variant in the cancer vs. the germline sample (Cibulskis et al., 2013; Saunders et al., 2012; Wilm et al., 2012).

These statistical methods fundamentally require multiple independent observations (supporting reads) of the somatic variant at any given genomic location to distinguish true mutations from sequencing artifacts. Mutect (Cibulskis et al., 2013), a state-of-the-art low-allele frequency somatic mutation caller, subjects each SNV to Bayesian classifiers that assume that the SNV either results from sequencing noise or that the site contains a true cancer variant. A true cancer-related SNV call is made when the log-likelihood ratio from the two models strongly favors the true cancer Bayesian classifier. This "locus-centric" type of cancer mutation detection can be readily achieved through increased depth of sequencing - so long as the tumor sample contains a high proportion of tumor DNA. However, these methods are significantly challenged in the ctDNA setting where the VAF is expected to be well below 1%. For example, a decrease of VAF to 5% and sequencing depth to 10X resulted in a decreased in the sensitivity of Mutect to below 0.1 (Cibulskis et al., 2013; Saunders et al., 2012; Wilm et al., 2012). Thus, locus-centric mutation callers are unable to perform effective mutation calling in the ultra-low VAFs observed in low disease-burden cfDNA settings.

We reasoned that to tackle this challenge, we would need a novel mutation detection framework. Specifically, we would need methodology to accurately distinguish true somatic cancer mutations from sequencing artifacts, even in ultra low tumor fractions that preclude the presence of multiple supporting independent observations (reads) in any given genomic location. We propose a "read-centric" alternative approach, and developed a convolutional neural network classifier - Kittyhawk - trained to discriminate between individual sequencing reads containing sequencing artifacts and sequencing reads harboring somatic cancer mutations. We take advantage of the fact that both cancer mutations and sequencing errors are systemic and governed by distinct signatures that can be learned and used for efficient signal to noise discrimination (e.g., mutagenesis processes such as exposure to tobacco or UV light are enriched in specific sequence contexts; Alexandrov et al. (2013)) 0.01%-1%, as well as with cfDNA samples from patients with early stage lung cancer and an individual with non-malignant lung nodules as controls.

## 2 METHODS

### 2.1 TRAINING DATASET SELECTION

We aim to use a training scheme that allows us to both detect true somatic mutations with high sensitivity and reject candidate mutations caused by systemic sequencing artifacts. As a proof-of-principle, we applied this methodology to ctDNA detection of NSCLC. This is due to (i) significant clinical need in sensitive non-invasive detection methods in NSCLC, (ii) high mutation rate in NSCLC (>10 mutations/Mb), and (iii) distinctive tobacco-related mutational sequence context

Table 1: Complete Datasets and patient information

| Patient | Dataset type | Cancer type | Mutations (Tumor-PBMC) | Training Reads | Validation/Test reads |
|---------|--------------|-------------|------------------------|----------------|-----------------------|
| CA0045 | Train | NSCLC | 12991 | 819200 | 59391 |
| CA0046 | Train | NSCLC | 12559 | 716800 | 80896 |
| CA0047 | Train | NSCLC | 18435 | 204800 | 11264 |
| CA0049 | Train | NSCLC | 9008 | 716800 | 50269 |
| CA0044 | Test / Synthetic | NSCLC | 8632 | Not used | 271360 |
| CA0035 | Train | Melanoma | 39158 | 1075200 | 102400 |
| CA0037 | Train | Melanoma | 98714 | 3123200 | 102400 |
| CA0038 | Train | Melanoma | 94850 | 5939200 | 102400 |
| CA0040 | Test / Synthetic | Melanoma | 59835 | Not used | 2816000 |
| BB1116 | cfDNA | NSCLC | 170894 | Not used | 16332191 |
| BB1125 | cfDNA | NSCLC | 7215 | Not used | 29321328 |
| BB672 | cfDNA | Healthy Donor | N/A | Not used | 0 |

signature (Alexandrov et al., 2013). We sampled four NSCLC patients and their whole-genome sequencing (Table 1) for tobacco-exposure lung cancer mutation learning, as well as their matched peripheral blood mononuclear cells (PBMC) germline DNA WGS for systematic sequencing artifact learning. To test our ability to extend this strategy to other cancer types, we also included three Melanoma patients to train a separate Melanoma-specific model. WGS libraries were prepared using the Illumina TruSeq Nano library preparation kit in accordance with the manufacturers instructions. Final libraries were quantified using the KAPA Library Quantification Kit (KAPA Biosystems), Qubit Fluorometer (Life Technologies) and Agilent 2100 BioAnalyzer, and were sequenced on an Illumina HiSeqX sequencer using 2 x 150bp cycles and processed in-house using our standard mutation calling pipeline (See appendix). Target depth of sequencing for both tumor and matched germline was greater than 40X. Next we curate all reads from these data that have either a true cancer mutation variant or a variant resulting from a sequencing artifact (see Figure 1 and attached appendix).

## 2.2 FEATURE CONSTRUCTION

To fully capture the sequencing read, alignment, and genomic context, we create a spatially-oriented representation of a read (Figure 2). Rows 1-5 represent the reference context (i.e., the corresponding sequence in the human genome, A,C,T,G and N for missing), while rows 6-10 represent the read sequence (A,C,T,G and N for missing). Rows 11-15 represent the information captured in the alignment string known as the CIGAR string (contains information about how each base aligned to the reference). We used the first five components of the CIGAR string, denoting a match or mismatch, an insertion or a deletion into the reference, a skipped position in the genome, and soft-clipped bases (positions which incur a modest penalty for being mismapped on the end of the read). The final row [16] represents the BQ score at each position in the read. Each column in our matrix represents an indicator vector, or one-hot encoding, referring to the presence or absence at a specific position along the read. For reads containing insertions in the reference, "N" is placed in the reference row at the location of the insertion to maintain the spatial alignment. For bases in the read that are deletions in the reference, "N" is instead placed in the read sequence at the location of the deletion.

The aligner may also implement a "soft masking" procedure to exclude a part of the read thought to have lost its sequencing accuracy (typically at the end of the read). Soft-masked regions of the read are modified such that consecutive Ns are inserted in the reference context rows. This is done to ensure the signal for softmasked regions is strong and to maintain the characteristic of these regions of independence from alignment. To ensure that the model is provided with adequate genomic context even if the variant appears at the ends of the read we add additional 25 bases of

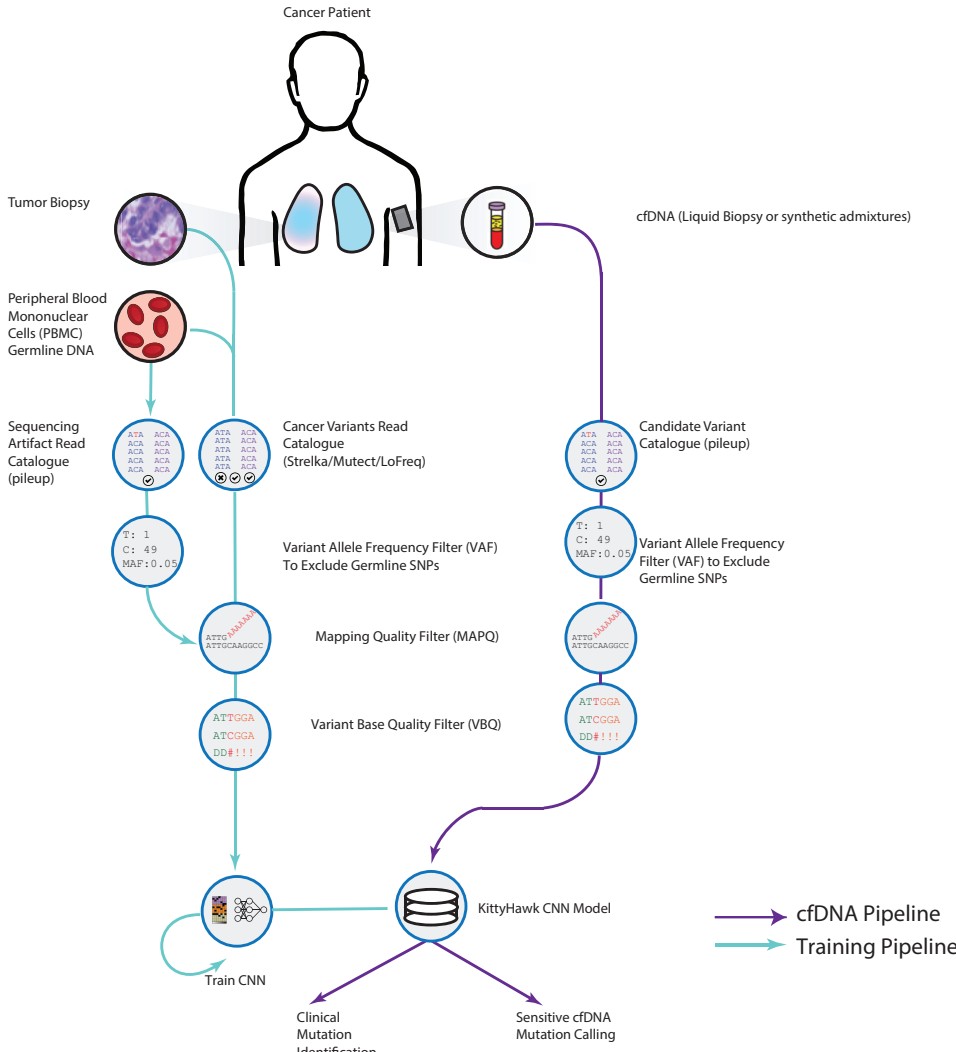

Figure 1: Kittyhawk Data processing pipeline, left (top to bottom): Whole-genome sequencing is performed using a tumor biopsy and PBMC, Variant and artifact catalogue are constructed using a consensus of Strelka, LoFreq, Mutect and a simple caller, pileup, respectively. Artifact reads are filtered such that only variants occurring once are retained. Mapping quality and variant base quality filters are applied to both datasets. Finally the datasets are balanced and the CNN is trained to convergence. Right (top to bottom): Both cfDNA and synthetic cfDNA admixtures have variant catalogues constructed using pileup, reads are then filtered by their variant allele frequency, then the mapping quality and variant base quality filters are applied. Reads then receive inference scores from Kittyhawk and may be used for downstream analysis.

genomic context to both ends of the read. This embodiment of the read results in a 16x200bp matrix for typical 150bp reads. To maintain equal read length, in the case where a read is shorter than 150bp, extra context bases are added. We set the maximum VBQ score at 40 (0.01% probability of sequencing error) and scale the scores to be in the interval [0, 1]. Bases not covered by a read (e.g., flanking genomic context) receive a base quality score of zero. Deletions in the read receive the mean BQ score of the two flanking positions.

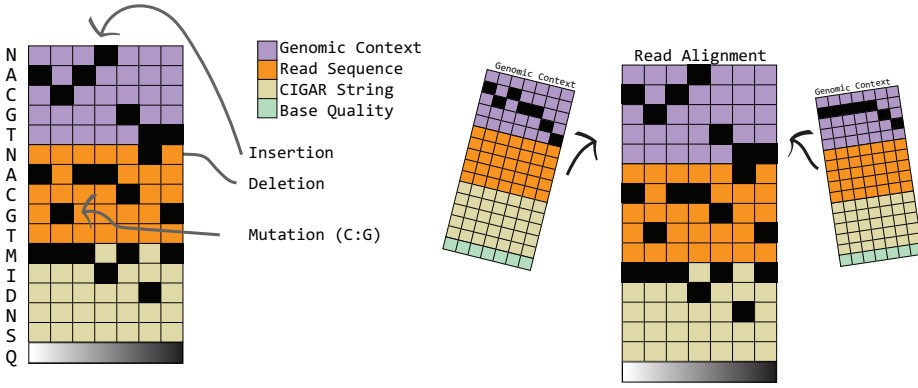

Figure 2: Top: Representation of a read and its alignment as seen by kittyhawk, Bottom: Genomic context is appended to the ends of the read. Zeroes are padded for non-context features.

## 2.3 MODEL, ARCHITECTURE, HYPER-PARAMETERS, AND IMPLEMENTATION

When designing a model for somatic mutation classification, it is important to recognize the sources of signal. A true mutation is likely to have a higher BQ regardless of its position in the read. Similarly, the read base, reference base, and alignment string (CIGAR) at the position of a true mutation are likely to be independent of the read alignment. More specifically, we can expect a true somatic mutation to be spatially invariant, while systemic errors in sequencing are strongly impacted by the position on the read. Nevertheless, some errors may have positional invariance. For example, sequencing artifacts caused by mis-mapping are likely to contain repetitive sequences or specific sequence motifs (such TTAGGG in telomeres). Our model must be able to accurately represent both the spatial invariance in true somatic mutations and in errors due to mapping, while simultaneously maintaining a model of (declining) BQ along the read. It follows that any shallow convolutional network that depends on a fully connected layer over the read of interest to make classifications would be unable to capture the invariance in the mutations. We elected for an 8-layer convolutional neural network with a single fully connected output layer inspired by the VGG architecture to correct for this spatial dependency (Simonyan & Zisserman, 2014). Building on the results of Alexandrov et al. (2013), which showed that tri-nucleotide context contains distinct signatures involved in mutagenesis, we convolve over all features (columns) at a position using a perceptive field of size three. After two successive convolutional layers, we apply down sampling by max-pooling with a receptive field of two and a stride of two, forcing our model to retain only the most important features in small spatial areas (Boureau et al., 2010). We propose two benefits from this architecture: (i) we maintain spatial invariance when convolving over trinucleotide windows and (ii) we can capture a "quality map" by collapsing the read fragment into 25 segments, each representing approximately an eight-nucleotide region.

The output of the last convolutional layer is applied directly to a sigmoid fully connected layer used to make the final classification. We use a simple logistic regression layer instead of a multi-layer perceptron or global average pooling to retain the features associated with position in the read. We deem our model, filters, and training scheme, Kittyhawk. Kittyhawk is the first use of a read representation that jointly captures the genomic context of alignment, the complete read sequence, and the integration of quality scores. Other works have used similar representations, but these consider piles of reads as single features, losing valuable information about the sequence alignment itself and the per-base quality associated with a read (Poplin et al., 2016; Torracinta & Campagne, 2016).

We trained our model using minibatch stochastic gradient decent with an initial learning rate $\lambda = 0.1$ and momentum $\gamma = 0.9$. The learning rate was decreased by a factor of 10 when the validation loss

reached a plateau as in He et al. (2016). We used a minibatch size of 256 as it provided a reasonable trade-off between validation loss and training speed. We elected to use a base of 64 filters, doubling after each max-pooling layer to maintain a consistent number of parameters at each convolutional layer. This was chosen empirically after observing the inability of a 32-base filter model to perform sufficiently on the lung cancer dataset. After each convolutional layer, we apply batch normalization (Ioffe & Szegedy, 2015) followed by a rectified linear unit (Nair & Hinton, 2010). Before each pooling layer, we apply dropout with a drop probability of 0.5 (Srivastava et al., 2014).

## 3 RESULTS

### 3.1 KITTYHAWK DISCRIMINATES SOMATIC MUTATIONS FROM NOISE WITH HIGH ACCURACY AND LEARNS CANCER-SPECIFIC MUTATIONAL SIGNATURES

To evaluate the performance of our model, we first examined the validation dataset comprised of 201,730 reads that were held out from training, from the four NCLSC patient samples used in model training (see section 2.1). This dataset includes 100,865 true mutated reads and 100,865 sequencing artifact containing reads that were not previously seen by the model. We evaluate our model with the metrics F1-score, precision, sensitivity, and specificity. We find that the CNN model provides an average of F1-score on the validation set of 0.961 (Table 2), comparable to methods that use multiple supporting reads for mutation calling (Poplin et al., 2016). Thus, the model is capable of accurately discriminating tumor cancer variant reads vs. sequencing artifact containing reads in a manner which is completely independent of depth of coverage and thus can be applied to any VAF.

To examine the generalizability of the model, we used it to analyze an additional NSCLC sample and its matched germline DNA, not used during model training (CA0044, Table 1). In this independent lung cancer case, we observe a F1 score of 0.92, confirming that the model is learning both lung cancer specific and sequencing artifact specific signatures for high accuracy discrimination (Table 2). To further examine this, we applied an additional sample from a patient with melanoma (CA0040, Table 1), which typically results in markedly distinct mutational profile due to the exposure to UV light instead of tobacco as the primary carcinogen (Figure 3a). Notably, our model achieves an F1-score of 0.71 on the melanoma sample. Thus, while the model is still sensitive, the lower precision and specificity in the melanoma sample indicate that Kittyhawk has learned specific mutation patterns associated with tobacco-exposed lung cancer, while learning a more general sequencing artifact pattern which is applicable to both tumor types.

To further explore this relationship, we measured the difference in tri-nucleotide context frequency (Figure 3b) between true cancer mutation variant reads and sequencing artifact containing reads from (i) lung cancer patient samples that were included in training (CA0046, validation dataset), (ii) lung cancer patient not included in training (CA0044), and (iii) the melanoma patient (CA0040). We note that as expected, the tobacco related lung adenocarcinoma samples show high enrichment in C>A transversions consistent with tobacco related mutational signature (Figure 3b). We hypothesized that Kittyhawk may learn specific sequence contexts that are prevalent in tumor mutational data (i.e., tumor-specific mutational signature). To test this hypothesis we measured the difference in frequency between true cancer variants vs. sequencing artifacts in each tri-nucleotide context, and correlated it with the average model prediction for these same reads. We reasoned that if the model is learning the (lung) cancer specific sequence context, we expect a high correlation between the tri-nucleotide sequence frequency and the model output. We found a high correlation between the model prediction and tri-nucleotide enrichment (Figure 3c), both in CA0046 (included in training, Pearsons r=1) and in CA0044 (not included in training, Pearsons r=0.95). Nevertheless, this high correlation may alternatively result from accurate classification that is independent of the sequence context. To directly examine this alternative scenario, we performed a similar analysis with the above described melanoma sample (CA0040). While we observe that a positive correlation (Pearsons r=0.64) between trinucleotide context and model predictions persists, indicating accurate classification derived from features other than the mutation signature alone, we also observe that this correlation is significantly lower than in the tobacco exposed lung cancer data. This finding is consistent with model learning of the specific lung cancer mutational signatures. This finding motivates us to train a separate model specifically geared towards detecting melanoma related so-

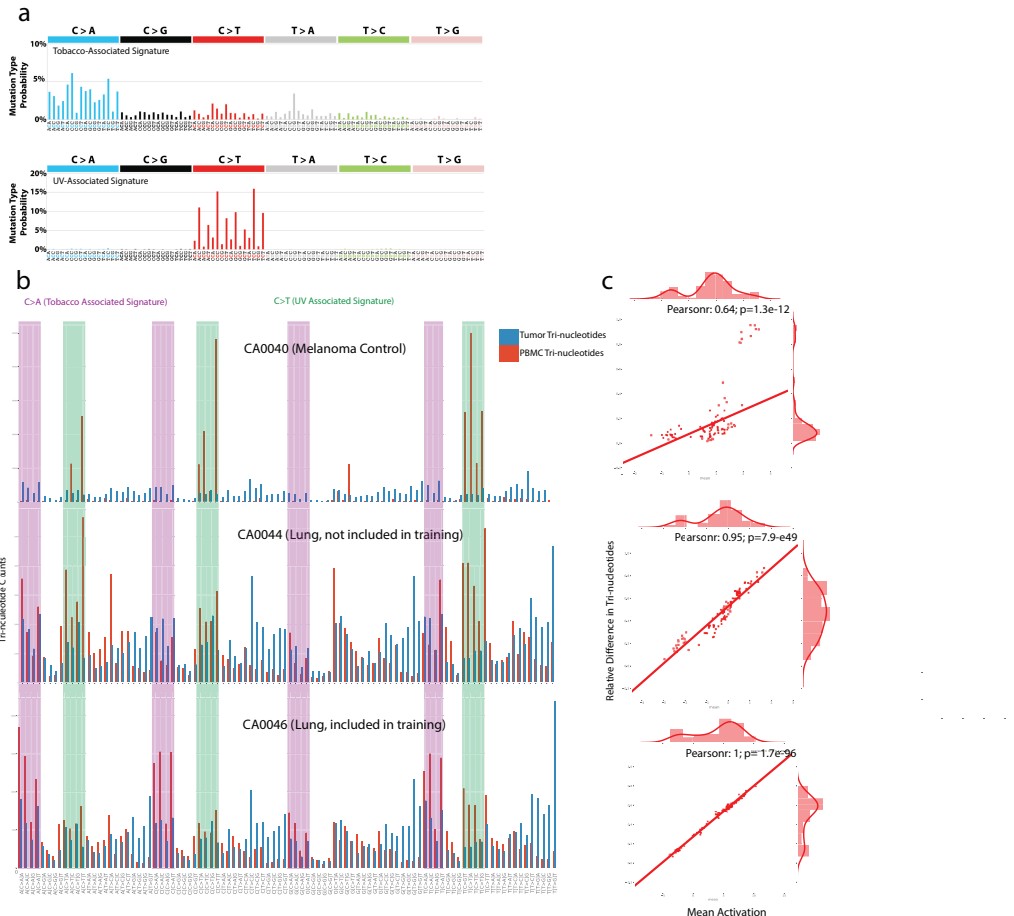

Figure 3: Kittyhawk Signature Analysis (from left): a) COSMIC signatures associated with Tobacco (top) and Melanoma (bottom) from Alexandrov et al. (2013). b) Tri-nucleotide frequencies from sample-specific Tumor and PBMC reads. Specific tri-nucleotides associated with Tobacco (purple) and UV radiation (green). c) Correlation of relative difference in tri-nucleotide frequencies and mean activiations of Kittyhawk.

matic mutations. We followed the same procedure described above for NSCLC, using an addition dataset from three melanoma patients. We observe similar performance, with high F1 score in the melanoma validation dataset, and independent melanoma sample, and a lower F1 score when this model was applied to NSCLC data (Table 3).

## 3.2 KITTYHAWK IS SENSITIVE AND PRECISE AT LOW TUMOR FRACTIONS IN SYNTHETIC PLASMAS

As noted above, the model performance is independent of coverage and VAF and thus is expected to perform well even in ultra-low VAF as seen in cfDNA of low disease-burden cancer. To directly test our method at low frequency mutation detection, we generated four simulated plasma samples from our test lung sample (CA0044, Table 1) by admixing randomly sampled reads from the patients matched germline DNA and from the patients tumor DNA. Sampling was performed to maintain depth of coverage of 35X and with tumor mixtures of 0, 1%, 0.1%, and 0.01%. Mixing was performed with three random seeds to generate independent replicates. While class labels provide which sample a given read originated from, tumor WGS may contain sequencing artifacts as well. We therefore undertook a more conservative approach and re-labeled true cancer variant reads as

Table 2: NSCLC Validation and Test Metrics

| Patient | Purpose | F1-score | Sensitivity | Specificity | Precision |
|---------|---------|----------|-------------|-------------|-----------|
| CA0045 | Train | 0.946 | 0.944 | 0.948 | 0.948 |
| CA0046 | Train | 0.962 | 0.949 | 0.976 | 0.975 |
| CA0047 | Train | 0.944 | 0.944 | 0.944 | 0.944 |
| CA0049 | Train | 0.976 | 0.975 | 0.978 | 0.977 |
| CA0044 | Test | 0.922 | 0.903 | 0.940 | 0.938 |
| CA0040 | Cancer Control | 0.718 | 0.793 | 0.642 | 0.689 |

any read from the tumor sample that also harbors a mutation from our tumor consensus mutation catalogue.

As anticipated, while the positive predictive value (PPV) decreases with lower tumor fraction in the admixture (reflecting the loss of true mutations due to the subsampling), the enrichment performance of Kittyhawk remains invariant across the range of tumor fractions, providing a 30X enrichment compared to the pileup method, a method which captures any observed mismatch, alone (Figure 4) (Li et al., 2009). We further compare this performance to several commonly used calling methods: Mutect, (a state of the art caller designed for low VAF mutation calling), Strelka (a somatic mutation caller with a similar design to Mutect), and SNooPer (a random forest mutation calling model), and demonstrate that unlike Kittyhawk, their performance rapidly drops when the tumor fraction decreases. Mutect and Strelka are unable to detect even a single mutation in the synthetic samples at any tumor fraction (VAF of 1% or less). SNooPer is only able to make mutation calls at tumor fraction of 1%, but not lower. It is important to note that this does not represent a failing of these cutting-edge tools, rather their distinct design purposes. All current mutation detection tools are designed to assess the information from multiple supporting reads to designate a genomic locus as mutated. In fact, we believe that in the settings for which these tools were designed, they likely outperform Kittyhawk, as the use of information from greater than one reads for mutation calling is expected to provide a more powerful classifier. However, as tumor derived genetic material is massively diluted in cfDNA, an alternative approach such as Kittyhawk is needed for effective filtering.

To evaluate that our method is robust in the clinical setting of cfDNA, we applied our approach to two patient derived cfDNA samples (Table 1), obtained at diagnosis. As a control, we obtained an additional cfDNA sample from an age matched individual with non-malignant lung nodule. We defined the compendium of true somatic mutation variants using the matched tumor and germline DNA obtained when these patients underwent subsequent surgical resection (as described in 2.1). The true positives are defined as reads classified by Kittyhawk as cancer variants which also overlap mutation calls derived from the traditional tumor and matched normal mutation calling. False positives were defined as reads which were classified as cancer variants by Kittyhawk and yet did not overlap the tumor/normal mutation catalogue. We note that this is a conservative definition, as cfDNA may show lesions not detected in the matched tumor due to spatial variation between mutated loci. Application of our approach on patient derived cfDNA recovered 114-132 somatic SNVs (sSNVs) out of 11,825-15,103 sSNVs detected in matched tumors consistent with a dilution of ctDNA to less than 1% in early stage cfDNA. Applying the same approach on a plasma sample from a patient with a benign lung nodule, show only 45-76 mutations that matched the same sSNV mutation compendium, indicating that Kittyhawk is able to detect relevant sSNV from plasma samples and discriminate malignant from benign samples. In parallel, Kittyhawk also suppressed the noise in the samples by filtering out 90-93% of reads with a variant. We note that analyzing the same samples through Mutect yield no variants detected for all of plasma samples.

## 4 DISCUSSION

Ultra-low tumor fraction such as observed in cfDNA fundamentally challenge the prevailing mutation calling paradigm. State-of-the-art mutation callers share a common unifying principle: mutation calling at a particular genomic location based on the observation of the cancer variant in multiple

Table 3: Melanoma Validation and Test Metrics

| Patient | Purpose | F1-score | Sensitivity | Specificity | Precision |
|---|---|---|---|---|---|
| CA0035 | Train | 0.925 | 0.927 | 0.925 | 0.924 |
| CA0037 | Train | 0.948 | 0.953 | 0.947 | 0.942 |
| CA0038 | Train | 0.935 | 0.931 | 0.935 | 0.938 |
| CA0040 | Test | 0.943 | 0.942 | 0.943 | 0.944 |
| CA0044 | Cancer Control | 0.721 | 0.589 | 0.764 | 0.928 |

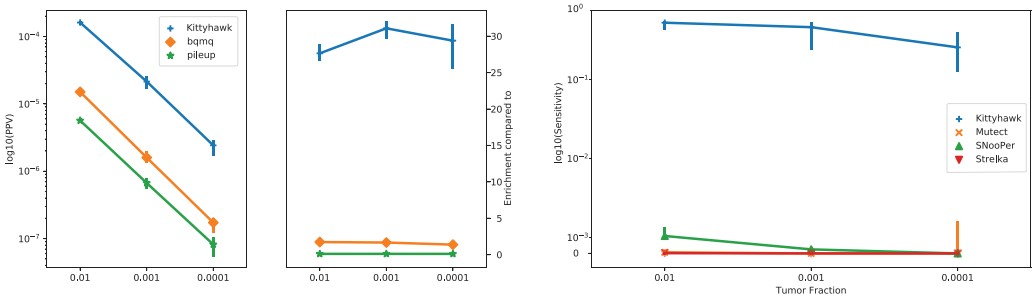

Figure 4: PPV, enrichment, and sensitivity of CA0044 synthetic cfDNA.

overlapping reads. However, in the ultra-low tumor fraction context, at best, only a single mutated read is observed, limiting the ability of traditional mutation calling.

The need for extending the mutation-calling framework to ultra-low tumor fraction contexts motivated us to rethink the mutation calling process from a locus-centric approach to a read-centric approach. This approach uses every individual read as input for a classifier and lends itself to the application of convolutional neuronal network learning. To realize this novel methodology, we embodied the information captured in the sequencing read (nucleotide sequence, context, quality metrics) in a spatial representation typically applied for image analysis. While we anticipate that our ongoing efforts to include larger training datasets, will result in further performance improvement, even at this proof-of-principle stage the algorithm is providing a 30-fold enrichment in a manner that is completely independent from variant allele fraction or depth of coverage, a unique performance feature that addresses a major emerging unmet need. Indeed, stable enrichment performance extends to tumor fractions as low as $10^{-4}$.

While Kittyhawk captures position in the read by using a fully connected sigmoid layer, there are other architectures, which may be suited for capturing relative position on the read. Additionally, we have excluded an extra source of information contained in the read-pair that comes from the DNA fragment. The read pair can be used to determine both the strand of origin (Watson or Crick) and to estimate the DNA fragment size. It has been observed that ctDNA have a distinct fragment size distribution compared to other cfDNA from normal cells (Underhill et al., 2016). It has been shown that recurrent neural networks (RNN) are a powerful tool for using length as a feature in bioinformatics at distances even up to 1kb, far beyond the size of a ctDNA fragment (Hill et al., 2017). These results suggest that integrating an RNN instead of a logistic regression layer could increase performance even further. In addition, while Kittyhawk was developed for the context of low tumor fraction mutation calling in cfDNA, we note that this framework can be adapted to other contexts. For example, it may be used in mutation (or germline SNP) detection in low pass genome sequencing (0.01-1X) across a wide range of applications. Furthermore, a read-centric approach may be also integrated with a more traditional locus-centric mutation calling approach, by adding Kittyhawk predictions as an additional input metric for extant statistical or machine learning mutation calling algorithms.

In summary, Kittyhawk is the first somatic mutation caller designed specifically to function in the ultra-low allele frequency setting where at best a single supporting read is available for candidate

mutation identification, such as liquid biopsy for early stage cancer detection. We apply a novel representation of a read together with a hand-engineered architecture to capture the entirety of informative features associated with a read and its alignment. This work sets the stage for a new family of somatic mutation callers to aid detection in liquid biopsy, paving the way for pivotal non-invasive screening and prognosis.

ACKNOWLEDGMENTS

A.Z. received support from EMBO long term fellowship (ALTF 140-2016)

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

# Appendix

GLOSSARY OF TERMS

**Base quality (BQ)**  – The massively parallel sequencing process provides an estimate of the confidence of the sequencing quality at each base, based on the sharpness of the optical read-out during sequencing. **Cell free DNA (cfDNA)** – Fragments of DNA are found within the plasma and are amenable to sequencing for clinical purposes in cancer and other applications. **CIGAR string** – a standardized output of genome aligners that provide alignment information compared with the matched reference sequence, including insertions, deletions and mismatches. This output takes the form of a string of indicators corresponding to the sequence alignment at each base. **Circulating tumor DNA (ctDNA)** – Dying tumor cells shed fragments of their DNA into the plasma and can be detected with cfDNA. **Mapping quality (MAPQ)** – Genomic aligners, such as BWA (Burrows-wheeler aligner), provide a confidence estimate regarding the accuracy of the mapping for each read. **Somatic cancer mutation** – a change in DNA that affects the cancer cells but not the germline of the patient. **Variant allele frequency (VAF)** – the frequency of the variant allele of the total number of alleles in the population. In the context of cancer sequencing this reflects the purity (i.e., fraction of DNA derived from tumor vs. normal cells) and ploidy (i.e. number of copies of each allele). For example, a heterozygous somatic mutation in diploid region of the genome affecting 50% of the cells in the sample is expected to have a VAF of 25% (50/100[cells] * 1/ 2[copies] = 0.25).

ADDITIONAL DETAIL REGARDING TRAINING DATA PROCESSING

We aim to use a training scheme that allows us to both detect true somatic mutations with high sensitivity and reject candidate mutations caused by systemic sequencing artifacts. As a proof-of-principle, we applied this methodology to ctDNA detection of NSCLC. This is due to (i) significant clinical need in sensitive non-invasive detection methods in NSCLC, (ii) high mutation rate in NSCLC (>10 mutations/Mb), and (iii) distinctive tobacco-related mutational sequence context signature (Alexandrov et al, 2013). We sampled four NSCLC WGS (**Table 1**) for tobacco-exposure lung cancer mutation learning, as well as their matched peripheral blood mononuclear cells (PBMC) germline DNA WGS for systematic sequencing artifact learning. To test our ability to extend this strategy to other cancer types, we also included three Melanoma patients to train a separate Melanoma-specific model. WGS libraries were prepared using the Illumina TruSeq Nano library preparation kit in accordance with the manufacturer's instructions. Final libraries were quantified using the KAPA Library Quantification Kit (KAPA Biosystems), Qubit Fluorometer (Life Technologies) and Agilent 2100 BioAnalyzer, and were sequenced on an Illumina HiSeqX sequencer using 2 x 150bp cycles and processed in-house using our standard mutation calling pipeline (See below). Target depth of sequencing for both tumor and matched germline was > 40X.

Next we curate all reads from these data that have either a true cancer mutation variant or a variant resulting from a sequencing artifact.

The true cancer variant reads were obtained through the following procedure (**Figure 1**):

> - Applying a consensus of the three leading mutation callers, Strelka, LoFreq, and Mutect (Saunders et al., 2012; Wilm et al., 2012; Cibulskis et al., 2013) to generate a catalogue of genomic loci with true somatic mutation SNVs.

> - We collect all reads supporting these mutations (~10-15 per site, considering heterozygous mutations, 40X sequencing depth and typical sample purity) and label them as true cancer mutation variant containing reads.

To enable model learning for discrimination against sequencing artifacts, we curate reads containing sequencing artifact variants through the following procedure:

> - Using the pileup method (a method which captures any observed mismatch [Li et al., 2009]), we identify all loci that contain a variant in the germline DNA samples from these patients. As this DNA is derived from non-malignant cells (PBMCs), we may assume that variants result primarily from sequencing artifacts.

> - We only retain variants supported by a single read in the 40X WGS data of these germline DNA samples. This step is added to exclude rare germline single nucleotide polymorphisms (SNPs). Intersection of these variants with the database dbSNP (build ID: 150) showed that this strategy is sufficient to filter out germline SNPs, with <0.1% overlapping with known SNPs.

Finally, we apply an additional filtering step to all reads (cancer mutation variant reads and sequencing artifact reads) to filter out read of poor quality that are overwhelmingly sequencing artifacts. Specifically, we filter out variants with a base quality (VBQ) score at the mutation less than 20. We selected this threshold based on the likelihood of error provided by the Illumina platform. VBQ scores correspond to $\log\_10(-q/10)$ probability of error, therefore more than 1 in

100 variants with a base quality below 20 represent a sequencing error. We then filtered reads where the mapping quality (MAPQ), or the likelihood that a given alignment is correct, below 40. This cutoff was chosen as mapping qualities from mapping software are bimodally distributed with modes at MAPQ=30 and MAP=60. We note that 201,730  (5-10%) reads per patient are held out during training and used to monitor training progress and to verify the model's performance on independent reads [*validation dataset*].

## Preprocessing

Before calling, tumor and matched normal DNA sequencing data go through our somatic pre-processing pipeline which includes aligning reads to the GRCh37 human reference genome using the Burrows-Wheeler Aligner (BWA) aln (Li and Durbin, 2009), marking of duplicate reads by the use of NovoSort ( a multi-threaded bam sort/merge tool by Novocraft technologies http://www.novocraft.com); realignment around indels (done jointly for all samples derived from one individual, e.g. tumor and matched normal samples, or normal, primary and metastatic tumor trios) and base recalibration via Genome Analysis Toolkit (GATK) (McKenna *et al.*, 2010).

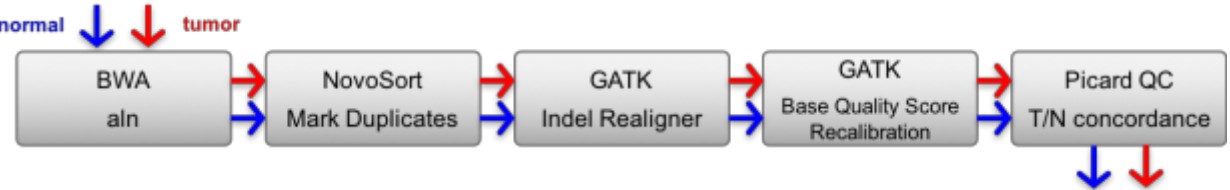

Figure 1: NYGC pre-processing pipeline.

## Quality control

**Basic DNA sequencing metrics.** We run a battery of Picard (QualityScoreDistribution, MeanQualityByCycle, CollectBaseDistributionByCycle, CollectAlignmentSummaryMetrics, CollectInsertSizeMetrics, CollectGcBiasMetrics, CollectOxoGMetrics) and GATK (FlagStat, ErrorRatePerCycle) metrics on all DNA data. In addition, for WGS experiments we run bedToolsCoverage and custom R scripts to compute sequencing depth of coverage, and for exomes and panels we run GATK CalculateHsMetrics and DepthOfCoverage modules. We perform outlier detection to identify samples that need to be manually reviewed, and if verified not to pass QC, failed.

**Sample contamination and tumor-normal concordance.** We run Conpair (Bergmann *et al.*, 2016) on all tumor-normal pairs to to detect cross-individual contamination and sample mix-ups.

**Autocorrelation.** We compute a metric called Autocorrelation (*Zhang et al.*, 2013) to give us an indication of unevenness in coverage in sequencing data. This method was originally developed for array data but we have adapted it for WGS data. We generate intervals with window size of 1kb every 10kb along the genome, calculate read depth in these windows using Picard HsMetrics and then compute Autocorrelation.

## Calling SNVs and indels

We return the union of somatic SNVs called by muTect (Cibulskis *et al.*, 2013), Strelka (Saunders *et al.*, 2012) and LoFreq (Wilm *et al.*, 2012) and the union of indels called by Strelka, and somatic versions of Pindel (Ye *et al.*, 2009) and Scalpel (Narzisi *et al.*, 2014).

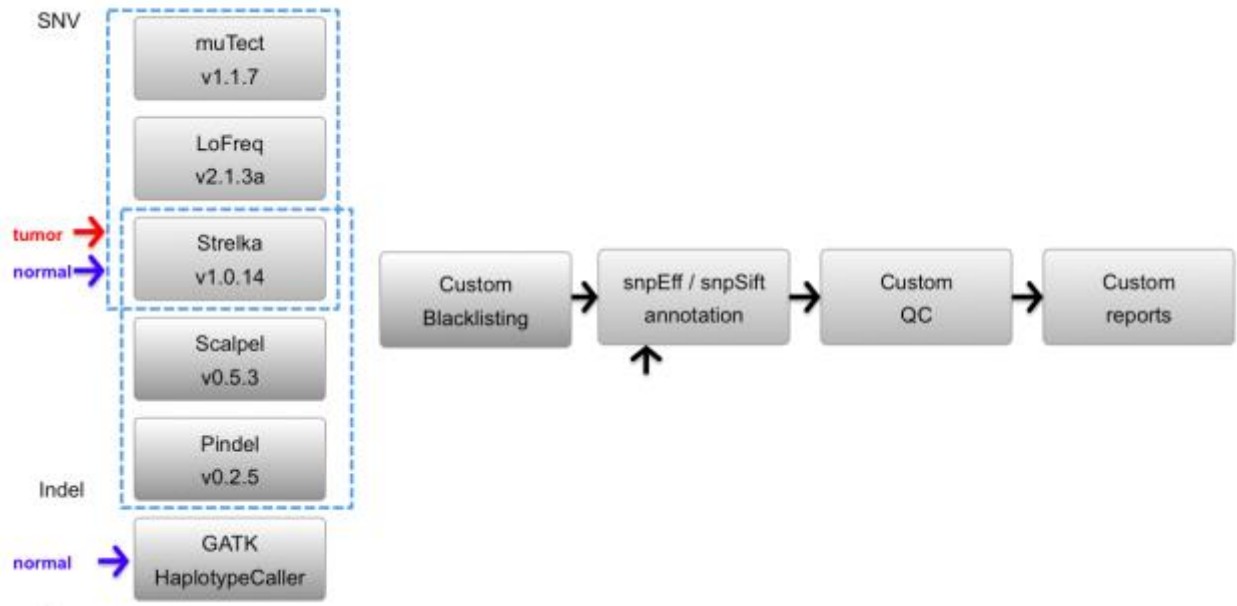

Figure 2: NYGC somatic SNV/indel pipeline.

The choice of SNV callers was based on internal benchmarking of individual and combinations of callers on a synthetic virtual tumor created by spiking reads from two HapMap samples in a way that mimics somatic variants with predefined variant allele frequencies (Cibulskis *et al.*, 2013). The choice of indel callers was based on internal benchmarking on synthetic data from the DREAM challenge (Ewing *et al.*, 2015).

For human samples, we also return germline calls in a panel of cancer risk genes (APC, ATM, BARD1, BMPR1A, BRCA1/2, BRIP1, CDH1, CDK4, CDKN2A, CHEK2, CYLD, EPCAM, IDH1/2, MEN1, MET, MLH1, MSH2/6, MUTYH, NBN, NF1/2, PALB2, PMS1/2, PRKAR1A, PTCH1, PTEN, RAD51C/D, RB1, RET, SDHAF2, SDHB/C/D, SMAD4, STK11, TP53, TSC1/2, VHL, WRN, WT1), made by the use of GATK HaplotypeCaller.

## Calling CNVs and SVs [WGS data only]

Structural variants (SVs), such as deletions and amplifications as well as copy-neutral genomic rearrangements are detected by the use of multiple tools (NBIC-seq (*Xi et al.*, 2016), Crest (*Wang et al.*, 2011), Delly (*Rausch et al.*, 2012), BreakDancer (*Chen et al.*, 2009)) that employ complementary detection strategies, such as inspecting read depth within genomic windows, analyzing discordant read pairs, and identifying breakpoint-spanning split reads.

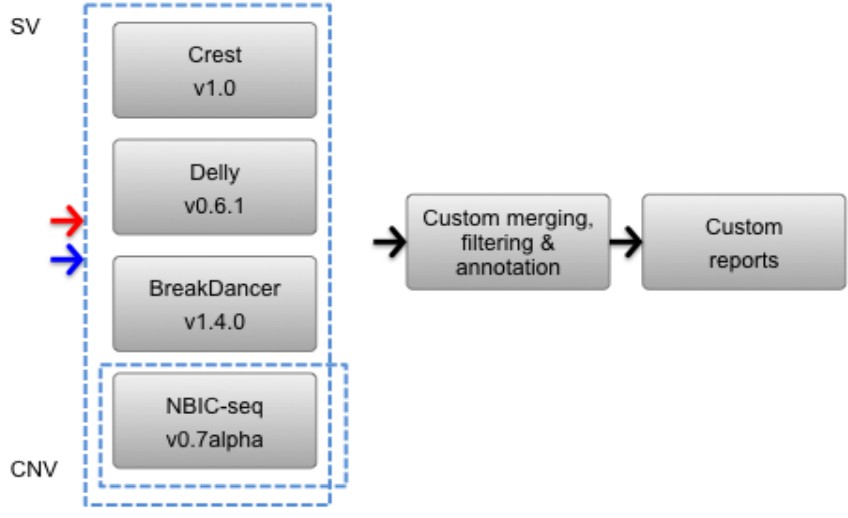

Figure 3: NYGC somatic WGS CNV/SV pipeline.

## Calling CNVs [WES data only]

We use EXCAVATOR (*Magi et al.*, 2013), a read depth based tool, to detect copy-number variants (CNVs) such as deletions and amplifications.

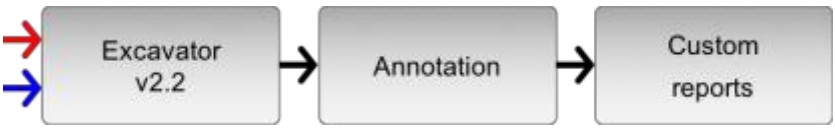

Figure 4: NYGC somatic WES CNV pipeline.

## Calling variants without a matched normal [human samples only]

When a matched normal sample is not available, in its place we use a "contemporary normal", that is, DNA from the HapMap sample NA12878 that was prepped and sequenced using the same protocol as the tumor sample. Using a contemporary normal removes some of the false positives that are due to prep and sequencing (that would manifest in the same way in the tumor and NA12878), as well as (mostly common) germline variants that are common to the tumor sample and NA12878.

## Processing of patient-derived xenograft (PDX) samples

PDX samples undergo an additional preprocessing step. Prior to the preprocessing pipeline, mouse reads are detected and removed by aligning the data to a combined reference genome of mouse (GRCm38/mm10) and human (GRCh37). All read pairs with both reads mapping to mouse or one read mapping to mouse and one unmapped are excluded from the subsequent processing and analyses steps.

## Filtering SNVs and indels

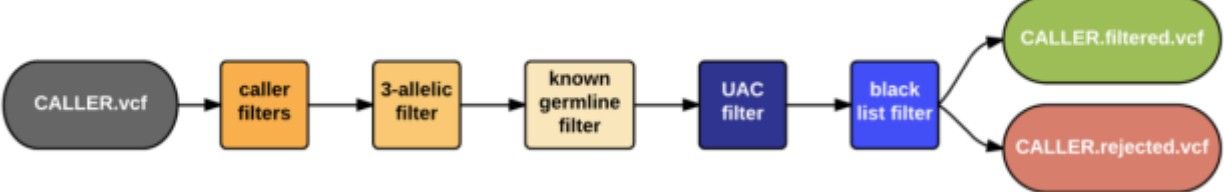

We use a multi-step filtering process:

Figure 5: The NYGC custom multi-step SNV/indel filtering

**Default caller filters.** SNVs and indels are filtered using the default filtering criteria as natively implemented in each of the callers. For Pindel and Scalpel (natively germline callers) we use custom in-house scripts for filtering. For each caller we keep these variants:

- LoFreq: FILTER=PASS
- muTect: variants with "PASS" in the filter field of the VCF file, which is equivalent to "KEEP" in the text file
- Strelka: FILTER=PASS
- Pindel: FILTER=PASS
- Scalpel: FILTER=PASS

**Triallelic positions.** The latest revision of the pipeline removes triallelic positions. Some SNV callers (e.g. muTect) remove them by default, and our internal investigation showed that triallelic sites within a sample are by and large due to an unmatched normal, not observing the second allele in the normal because of low coverage in the normal at that locus, or mapping artifacts.

**Common germline variants.**
Human samples:

The resulting set of SNVs and indels is further filtered with common variants seen at

MAF ≥ 5% in DNMT3A, TET2, JAK2, ASXL1, TP53, GNAS, PPM1D, BCORL1 and

SF3B1 genes (see Xie *et al.*, 2014) and with MAF ≥ 1% elsewhere in the genome, as

reported in the 1000 Genomes Project release 3 (1000 Genomes Project Consortium,

2012) and the Exome Aggregation Consortium (ExAC) server

(http://exac.broadinstitute.org), because these are very unlikely to be important in

cancer.
Mouse samples:

The resulting set of SNVs and indels is further filtered with variants seen in dbSNPv138
and Mouse Genome Project (v3).

**UAC filter.** Because callers often return different ref/alt allele counts for the same variant we introduced unified allele counts (UAC). Computation of UAC is based on the bam-readcount tool (Larson *et al.*, 2012). For each variant we generate 4 values that are independent of callers:

tumor-ref, tumor-alt, normal-ref, normal-alt. If the tumor_VAF < normal_VAF we discard the variant.

**Artifacts [human samples only].** In addition, we remove a subset of artifactual calls by the use of an blacklist created by calling somatic variants on 16 random pairings of 80x/40x in-house sequenced HapMap WGS data.

**More.** If you wish to further filter the variant call set, the bam-readcount tool (https://github.com/genome/bam-readcount) will provide a list of technical co-variates (eg. mapping or base quality statistics) for each position in the tumor and normal BAM files.

### Annotation and prioritization of SNVs and indels
Human samples:

Variants are annotated for their effect (non-synonymous coding, nonsense, etc.) using snpEff (Cingolani *et al.*, 2012) based on human genome annotations from ENSEMBL. We further annotate the variants via snpEff, snpSift and GATK VariantAnnotator module with information from COSMIC (Forbes *et al.*, 2012), 1000 Genomes Project, ExAC, CIViC (Clinical Interpretation of Variants in Cancer, https://civic.genome.wustl.edu), UniProt (http://www.uniprot.org), etc. We return variant prioritization scores for coding changes based on CHASM (Carter *et al.,* 2009), MutationAssessor (Reva *et al.,* 2011) and FATHMM Somatic (Shihab *et al.,* 2013).

Mouse samples:

Variants are annotated for their effect (non-synonymous coding, nonsense, etc.) using snpEff (Cingolani *et al.*, 2012) based on mouse genome annotations from ENSEMBL.

### Filtering and annotation of SVs and CNVs [WGS data only]

All filtering and annotation of SVs and CNVs is done with in-house scripts, making heavy use of bedtools (http://bedtools.readthedocs.org).

**SV merging.** We merge and annotate SVs called by Crest, Delly and BreakDancer using BEDPE format. Two SV calls are merged if they share at least 50% reciprocal overlap (for intra-chromosomal SVs only), their predicted breakpoints are within 300bp of each other and breakpoint strand orientations match for both breakpoints. Thus, merging is done independent of which SV type was assigned by the SV caller (a classification that we found to be unreliable and variable from caller to caller).

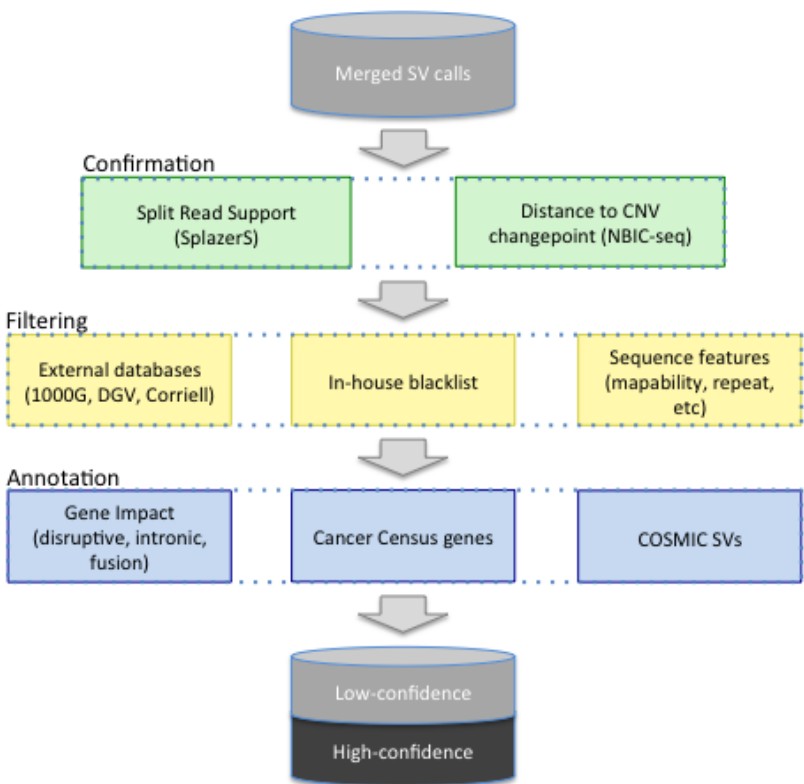

Figure 6: NYGC somatic CNV/SV filtering and annotation pipeline.

**Additional SV confirmation.** After merging, we annotate each SV with the closest CNV changepoint as detected by NBIC-seq from read depth signals. This adds confidence to true SV breakpoints that are not copy-neutral. Additionally, we do an independent sensitive split read check for each breakpoint using SplazerS. Apart from adding confidence and basepair precision to the breakpoint, this step also helps remove remaining germline SVs also found in the normal.

**SV filtering.** Some SV callers still suffer from large numbers of false positives; those are often due to germline SVs overlooked in the normal, e.g. because of low coverage or an unmatched normal, or systematic artifacts due to mapping ambiguities. We annotate and filter germline variants through overlap with known SVs (1000G call set, DGV for human; MGP for mouse) as well as through overlap with an in-house blacklist of SVs (germline SVs and artifacts called in healthy genomes). As mentioned above, also the split read check helps remove remaining germline SVs.
Finally, we prioritize SVs that were called by more than one tool, or called by only one tool but also confirmed by 1) a CNV changepoint, or 2) at least 3 split reads (in tumor only). Since we found them to be very specific, we also keep Crest-only calls in the high confidence set.

**SV/CNV Annotation.** All predicted copy number and structural variants are annotated with gene overlap (RefSeq, Cancer Census) and potential effect on gene structure (e.g. disruptive, intronic, intergenic). If a predicted SV disrupts two genes and strand orientations are compatible, the SV is annotated as a putative gene fusion candidate. Note that we do not check reading frame at this point. Further annotations include sequence features within breakpoint flanking regions, e.g. mappability, simple repeat content, segmental duplications and Alu repeats.

## Filtering and annotation of CNVs [WES data only]

All filtering and annotation of CNVs is done with in-house scripts, making heavy use of bedtools (http://bedtools.readthedocs.org).

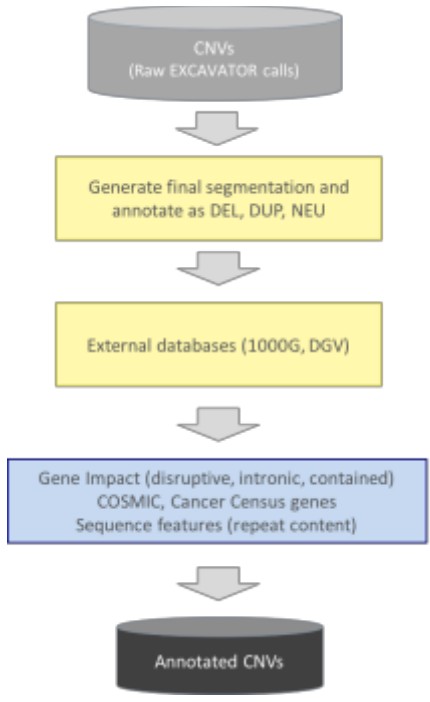

Figure 6: NYGC somatic CNV annotation pipeline.

**Final Segmentation.** Adjacent targets (intervals) from the same chromosome and having the same normalized mean read count are merged together to generate the final segmentation and further annotated as deletion, amplification or copy-neutral.

**Annotation.** All predicted CNVs are annotated with germline variants through overlap with known events (1000G call set, DGV for human). Cancer-specific annotation includes overlap with genes (RefSeq, Cancer Census) and potential effect on gene structure (e.g. disruptive, intronic, intergenic). Sequence features within breakpoint flanking regions, e.g. mappability, simple repeat content, segmental duplications and Alu repeats are also annotated. CNVs of size <20Mb are denoted as focal and the rest are large-scale.

## Delivered files

We return the caller-ready BAM files (*.final.bam) for the tumor and matched normal sample.

**SNVs/indels.** The SNV/indel pipeline returns the raw outputs of all variant callers, in VCF format (and for muTect also in TXT format).

We in addition return the annotated union of all SNVs (*.snv.union.v*.*), union of all indels (*.indel.union.v*.*), and union of all SNVs and indels together (*.union.v*.*), in three formats:

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
