# OpenReview forum: "Deep learning mutation prediction enables early stage lung cancer detection in liquid biopsy"
_ICLR.cc/2018/Conference — Invite to Workshop Track_

### Official Review · AnonReviewer1 · 2017-11-26
**Well written paper, presenting an effective DL solution to an important computational biology problem**

**Rating:** 8
**Confidence:** 4

**Review:**

In this paper the author propose a CNN based solution for somatic mutation calling at ultra low allele frequencies.
The tackled problem is a hard task in computational biology, and the proposed solution Kittyhawk, although designed with very standard ingredients (several layers of CNN inspired to the VGG structure), seems to be very effective on both the shown datasets.
The paper is well written (up to a few misprints), the introduction and the biological background very accurate (although a bit technical for the broader audience) and the bibliography reasonably complete. Maybe the manuscript part with the definition of the accuracy measures may be skipped. Moreover, the authors themselves suggest how to proceed along this line of research with further improvements.
I would only suggest to expand the experimental section with further (real) examples to strengthen the claim.
Overall, I rate this manuscript in the top 50% of the accepted papers.

---

> ### Author Response · Authors · 2018-01-05
> **Response**
>
> We thank the reviewer for appreciation of the challenges of this task as well as its importance. We have revised the manuscript to increase its clarity for a broader audience and performed further proofing.  As suggested, we have further expanded the dataset.

---

### Official Review · AnonReviewer3 · 2017-11-27
**The paper offers a multi layer CNN to identify DNA sequence reads that capture cancer mutations and not sequencing errors.**

**Rating:** 4
**Confidence:** 3

**Review:**

Summary:

In this paper the authors offer a new algorithm to detect cancer mutations from sequencing cell free DNA (cfDNA). The idea is that in the sample being sequenced there would also be circulating tumor DNA (ctDNA) so such mutations could be captured in the sequencing reads. The issue is that the ctDNA are expected to be found with low abundance in such samples, and therefore are likely to be hit by few or even single reads. This makes the task of differentiating between sequencing errors and true variants due to ctDNA hard. The authors suggest to overcome this problem by training an algorithm that will identify the sequence context that characterize sequencing errors from true mutations. To this, they add channels based on low base quality, low mapping quality. The algorithm for learning the context of sequencing reads compared to true mutations is based on a multi layered CNN, with 2/3bp long filters to capture di and trinucleotide frequencies, and a fully connected layer to a softmax function at the top. The data is based on mutations in 4 patients with lung cancer for which they have a sample both directly from the tumor and from a healthy region. One more sample is used for testing and an additional cancer control which is not lung cancer is also used to evaluate performance.

Pros:

The paper tackles what seems to be both an important and challenging problem. We also liked the thoughtful construction of the network and way the reference, the read, the CIGAR and the base quality were all combined as multi channels to make the network learn the discriminative features of from the context. Using matched samples of tumor and normal from the patients is also a nice idea to mimic cfDNA data.

Cons:

While we liked both the challenge posed and the idea to solve it we found several major issues with the work.

First, the writing is far from clear. There are typos and errors all over at an unacceptable level. Many terms are not defined or defined after being introduced (e.g. CIGAR, MF, BQMQ). A more reasonable CS style of organization is to first introduce the methods/model and then the results, but somehow the authors flipped it and started with results first, lacking many definitions and experimental setup to make sense of those.  Yet Sec. 2 “Results” p. 3 is not really results but part of the methods. The “pipeline” is never well defined, only implicitly in p.7 top, and then it is hard to relate the various figures/tables to bottom line results (having the labels wrong does not help that).

The filters by themselves seem trivial and as such do not offer much novelty. Moreover, the authors filter the “normal” samples using those (p.7 top), which makes the entire exercise a possible circular argument.

If the entire point is to classify mutations versus errors it would make sense to combine their read based calls from multiple reads per mutations (if more than a single read for that mutation is available) - but the authors do not discuss/try that.

The entire dataset is based on 4 patients. It is not clear what is the source of the other cancer control case. The authors claim the reduced performance show they are learning lung cancer-specific context. What evidence do they have for that? Can they show a context they learned and make sense of it? How does this relate to the original papers they cite to motivate this direction (Alexandrov 2013)? Since we know nothing about all these samples it may very well be that that are learning technical artifacts related to their specific batch of 4 patients. As such, this may have very little relevance for the actual problem of cfDNA.

Finally, performance itself did not seem to improve significantly compared to previous methods/simple filters, and the novelty in terms of ML and insights about learning representations seemed limited.

Albeit the above caveats, we iterate the paper offers a nice construction for an important problem. We believe the method and paper could potentially be improved and make a good fit for a future bioinformatics focused meeting such as ISMB/RECOMB.

---

> ### Author Response · Authors · 2018-01-05
> **Point-by-point response - first half**
>
> We thank the reviewer for the careful examination and critique. We have tried to address all concerns as detailed below and hope that the manuscript is now significantly improved.
> Point-by-point response:
> 1. "First, the writing is far from clear. There are typos and errors all over at an unacceptable level."
> We regret that these errors have been included in the submission. In our revised manuscript we have performed a more rigorous proofing and hope to have resolved this issue.
> 2." Many terms are not defined or defined after being introduced (e.g. CIGAR, MF, BQMQ)."
> We thank the reviewer for this important comment and have included clear definitions as well as a glossary of terms used to enhance the readability of the manuscript to diverse audiences.
> 3. "A more reasonable CS style of organization is to first introduce the methods/model and then the results, but somehow the authors flipped it and started with results first, lacking many definitions and experimental setup to make sense of those. Yet Sec. 2 “Results” p. 3 is not really results but part of the methods."
> We have revised the manuscript as suggested to address this concern and follow a CS format.
>  4. "The “pipeline” is never well defined, only implicitly in p.7 top, and then it is hard to relate the various figures/tables to bottom line results (having the labels wrong does not help that)."
> We have included an appendix detailing the mutation calling pipeline and a detailed figure (Figure 1) dedicated to provide an overview of the entire procedure. We have corrected the labeling issues.
> 5. "The filters by themselves seem trivial and as such do not offer much novelty."
> The filters we have used first include rejecting reads with very low base quality and mapping quality. This filtering step allows the CNN to learn more complex features and interactions between the features.  In the application of Kittyhawk to cfDNA, we apply an additional filter on variant allele frequency to exclude private germline single nucleotide polymorphisms, allowing the direct application of this algorithm even in the absence of matched normal germline DNA.
> 6. "Moreover, the authors filter the “normal” samples using those (p.7 top), which makes the entire exercise a possible circular argument. "
> Indeed, low base and mapping quality germline DNA reads (“normal” samples) were filtered prior to the use of the reads in model training. As noted above, this was done in order to allow the CNN to learn more complex features that distinguish true mutated reads and artifactually altered reads. In the implementation of our strategy to either synthetic or real cfDNA data we also include this first filtering step to remove these reads, as we have shown them to be highly enriched in sequencing artifacts. We note no circularity in this approach, as the cfDNA data includes no labeling of normal DNA vs. tumor DNA reads.
> 7. "If the entire point is to classify mutations versus errors it would make sense to combine their read based calls from multiple reads per mutations (if more than a single read for that mutation is available) - but the authors do not discuss/try that. "
> We are grateful for this suggestion. Indeed future integration of our model with extant mutation caller can be considered, to improve mutation calling in the setting of multiple supporting reads per mutated locus. We have not developed this aspect as it is not directly related to the unique challenge we are tackling (variant allele fraction far lower than depth of sequencing). We have included a discussion of such a potential integration.

---

> > ### Author Response · Authors · 2018-01-05
> > **point-by-point response- second half**
> >
> > 8. "The entire dataset is based on 4 patients. It is not clear what is the source of the other cancer control case."
> >
> > We regret the lack of clarity on our part in the original manuscript. We have used data from 5 patients with lung cancer, 4 patients with melanoma as well as 2 early lung cancer cfDNA and a cfDNA control from a patient with a benign lung nodule. These are now detailed in Table 1.  We are currently expanding this dataset using a WGS dataset of more than a 100 NSCLC patients, which we anticipate to complete prior to the presentation of the work. The nature of controls used in all of the analyses have been clarified.
> >
> > 9. "The authors claim the reduced performance show they are learning lung cancer-specific context. What evidence do they have for that? Can they show a context they learned and make sense of it? How does this relate to the original papers they cite to motivate this direction (Alexandrov 2013)? "
> >
> > We thank the reviewer for this important comment. We have added an additional figure addressing this issue (Figure 3). Namely, to assess the ability of the model to detect specific mutational signatures, we have measured the difference in the tri-nucleotide distributions between true cancer variants and sequencing artifact variants, and correlated the models score with these differences. We found a strong correlation that is specific to lung cancer samples, and less so to melanoma. Furthermore, we show that a model developed for lung cancer underperforms in melanoma and vice versa.
> >
> > 10. "Since we know nothing about all these samples it may very well be that that are learning technical artifacts related to their specific batch of 4 patients. As such, this may have very little relevance for the actual problem of cfDNA."
> >
> > As shown in Figure 3 and as described above, the model is learning sequence contexts that are generally observed in lung adenocarcinoma rather than specific to this batch of patients.  Furthermore, to demonstrate the applicability of this method to cfDNA, we tested it on patient-derived cfDNA.  The robust performance in both of these settings suggests that the Kittyhawk model is indeed learning more general features that define true tumor variants vs. sequencing artifacts, rather than specific batch characteristics.
> >
> > 11. "Finally, performance itself did not seem to improve significantly compared to previous methods/simple filters, and the novelty in terms of ML and insights about learning representations seemed limited"
> >
> > We respectfully disagree. To our knowledge, Kittyhawk is the first tool to directly address the challenge of mutation calling in the setting of variant allele frequency lower than the depth of coverage, a major emerging challenge as noted by all reviewers. For comparison, Mutect, a state-of-the-art caller delivers no mutation calls at a tumor fraction of 1:1000.  The ability to tackle ultra-low frequency mutations is done through a reframing of the mutation-calling problem from a locus-centric approach to a read-centric approach. This reframing is empowered by the embodiment of the read as features amenable to CNN learning originally designed for image learning. As such, it serves to extend the application of CNN to an important clinical area of development. While we anticipate that our ongoing efforts, that include larger datasets in training, will result in further performance improvement, even at this proof-of-principle stage the algorithm is providing a 30-fold enrichment. Notably, this is done in a manner that is completely independent from variant allele fraction, a unique performance feature that addresses a major emerging unmet need.

---

### Official Review · AnonReviewer2 · 2017-11-28
**Interesting method applied to an important problem. Would like to see an inclusion of more recent methods.**

**Rating:** 5
**Confidence:** 4

**Review:**

his paper proposes a deep learning framework to predict somatic mutations at extremely low frequencies which occurs in detecting tumor from cell-free DNA. They key innovation is a convolutional architecture that represents the invariance around the target base. The method is validated on simulations as well as in cfDNA and is s
hown to provide increased precision over competing methods.

While the method is of interest, there are more recent mutation callers that should be compared. For example, Snooper which uses a RandomForest  (https://bmcgenomics.biomedcentral.com/articles/10.1186/s12864-016-3281-2) and hence would be of interest as another machine learning framework. They also should compare to Strelka whic
h interestingly they included only to make final calls of mutations but not in the comparison.

Further, I  would also have liked to see the use of standard benchmark datasets for mutation calling ( https://www.nature.com/articles/ncomms10001).

It appears that the proposed method (Kittyhawk) has a steep decrease in PPV and enrichment for low tumor fraction which are presumably the parameter of greatest interest. The authors should explore this behavior in greater detail.

---

> ### Author Response · Authors · 2018-01-05
> **Note on differences from other available callers as well as performance in low tumor fraction.**
>
> We thank the reviewer for finding our submission of significance and acknowledging the importance of this problem.
>
> In our revision we have included the tools suggested in benchmarking (Snooper, Strelka), and have updated the results and figures. However, it is important to note, that our tool does not fulfill the same function as these tools and therefore direct benchmarking is not always informative. All current tools are designed to assess the information from multiple supporting reads to designate a genomic locus as mutated or unmutated. In fact, we believe that in the settings for which these tools were designed, they likely outperform Kittyhawk, as the use of information from >1 reads to make a mutation call is expected to provide a more powerful classifier.  Kittyhawk was designed to address a different context in which the variant allele frequency is far lower than the depth of sequencing (such as low tumor burden cfDNA), such that only one read is expected to support a mutation call at best. This setting required a conceptual rethinking of the problem where mutation calling is driven by classifying the individual read rather than a locus. While Snooper formally supports a single supporting read, in practice, the median variant allele frequency the authors have reported was 0.38, and given the average depth of sequencing used, most loci had >1 supporting reads.  This same issue limits also the use of benchmarking datasets suggested by the reviewer,  where the variant allele fraction is greater than the context of low tumor burden cfDNA. For this reason we generated simulated plasma to reflect these conditions and optimize our method for this setting.
>
> We also would like to thank you for emphasizing the importance of the performance in the lower tumor-fraction settings. The decrease in PPV with decreasing tumor fractions is expected even with stable sensitivity and specificity. This is due to fact that PPV is strongly affected by the prevalence of truly mutated reads in data.  Thank you for the observation about enrichment, we believe this was due to an error in the initial submission. Our revision now includes the corrected enrichment plots, showing stable enrichment across the tumor fraction range.

---

### Public Comment · (anonymous) · 2017-11-29
**Appendix and raw data**

Hello,

I would like to reproduce your results as part of the ICLR 2018 Reproducibility Challenge (http://www.cs.mcgill.ca/~jpineau/ICLR2018-ReproducibilityChallenge.html).

You mentioned in your paper an appendix which I don't seem to have access to; would you be able to make that available? I would also like to know where the raw data samples are available.
Would you be able to share Code with me?

Thank you very much

---

> ### Public Comment · (anonymous) · 2017-12-01
> **Appendix and raw data**
>
> If you cannot share your data with me, can you at least publish the appendix mentioned in the paper? perhaps I can get a similar dataset and still use the paper for the challenge.

---

> > ### Author Response · Authors · 2018-01-07
> > **Notes on reproducibility**
> >
> > Thank you for your interest in our project.  Unfortunately, we used clinical data for this manuscript and will be unable to share due to privacy concerns. The data is currently being deposited to a public repository with appropriate data access procedures, and we will share the access information once available. In the meanwhile, we would recommend accessing TCGA and using LUAD whole genome data. In our revision we have included detailed information about the entire process including pre-processing of WGS, read selection for training, and CNN parameter/architecture. We hope that this is sufficient to reproduce our results and we are happy to further facilitate in any way.

---

### Decision · Program_Chairs · 2018-01-29
**ICLR 2018 Conference Acceptance Decision**

**Decision:**

Invite to Workshop Track

**Comment:**

Authors present a method for representing DNA sequence reads as one-hot encoded vectors, with genomic context (expected original human sequence), read sequence, and CIGAR string (match operation encoding) concatenated as a single input into the framework. Method is developed on 5 lung cancer patients and 4 melanoma patients.


Pros:
- The approach to feature encoding and network construction for task seems new.
- The target task is important and may carry significant benefit for healthcare and disease screening.

Cons:
- The number of patients involved in the study is exceedingly small. Though many samples were drawn from these patients, pattern discovery may not be generalizable across larger populations. Though the difficulty in acquiring this type of data is noted.
- (Significant) Reviewer asked for use of public benchmark dataset, for which authors have declined to use since the benchmark was not targeted toward task of ultra-low VAFs. However, perhaps authors could have sourced genetic data from these recommended public repositories to create synthetic scenarios, which would enable the broader research community to directly compare against the methods presented here. The use of only private datasets is concerning regarding the future impact of this work.
- (Significant) The concatenation of the rows is slightly confusing. It is unclear why these were concatenated along the column dimension, rather than being input as multiple channels. This question doesn't seem to be addressed in the paper.
Given the pros and cons, the commitee recommends this interesting paper for workshop.